# Can visual impairment impact your income potential?

**Colinie Wickramaarachchi[1], Ruwan Jayathilaka◯[2]\*, Theekshana Suraweera[2], Samantha Thelijjagoda[2], Lakshika Kollure[1], Thushya Liyanage[1], Wageesha Serasinghe[1], Samanthi Bandara[2]**

**1** Department of Business Management, SLIIT Business School, Sri Lanka Institute of Information Technology, Malabe, Sri Lanka, **2** Department of Information Management, SLIIT Business School, Sri Lanka Institute of Information Technology, Malabe, Sri Lanka

\* ruwan.j@sliit.lk

**Data Availability Statement:** All relevant data are within the paper and its Supporting Information files.

**Funding:** This work was supported by the World Bank, Sri Lanka (AHEAD DOR Grant number 09).

## Abstract

People's quality of life (QOL) has been disrupted globally in the wake of the pandemic in recent times. This was mainly due to global economic crises fuelled by the coronavirus (COVID– 19) and other related factors. Sri Lanka, too, was facing major social and economic constraints in the period 2021–2022. Thus, all communities islandwide have been economically disturbed. Among others, people with Visual Impairment and Blindness (VIB) have been pushed to severely disadvantageous positions, financially and otherwise. A sample from three geographical locations in Sri Lanka; and eleven individuals representing diverse cadres in Sri Lankan society were purposely selected for the study based on the existence of the majority of the visually impaired community using a mixed approach. Descriptive statistics were utilised to analyse the identified socio-economic characteristics. Ordered probit regression was employed to determine the mediating effect of socio-economic status on income levels. Word Cloud illustrates the factors affecting the QOL. Most severely impaired individuals are more likely to earn a lower rate of income. This situation has degraded their lives and poor QOL. Participants' responses elucidate that facilities, resources, education, opportunities, income, employment, and government activities would enhance their QOL. The study adds value to society by recognising VIB people, helping them gain financial independence and strengthening them without marginalising the impaired community. The proposed policies in this study would be valuable for these social groups to address their wealth concerns.

## Introduction

The modern world consists of diverse demarcations and the level of personal income influences life standards as per social constructs. Grappling with challenging living conditions, one should be strong regarding economic independence. 'Health is wealth' is an adage popular among people that highlights the importance of physical and mental stability. At present, an individual's health benefits seem to be strongly associated with one's income and quality of life (QOL) [1]. A segment of society specifically subjected to economic disparity in terms of health

The funders had no role in study design, data collection and analysis, decision to publish, or preparation of the manuscript.

**Competing interests:** The authors have declared that no competing interests exist.

defects is identified as people with Visual Impairment and Blindness (VIB). Addressing the prevalence of people with visual disparities constitutes a sizeable portion of the public health challenge. The analysis unveils that regarding the global distribution of VIB, approximately 2.2 billion people are visually impaired, and almost 1 billion people with these disparities are curable with necessary preventive measures [2]. These visual impairments are considered bodily conditions that obstruct a person from living a good life in society [3]. The relationship between VIB and socio-economic characteristics has been discussed in many countries; in this context, some epidemiology studies have noted a significant relationship between VIB and socio-economic characteristics is particularly true in many rapidly growing countries, especially China [4]. Moreover, the global spread of VIB exemplifies that visual difficulties are generally more prevalent and severe in developing countries' rural areas, especially Sri Lanka. The survey conducted by the VISION-2020 national workshop programme by the Government of Sri Lanka (GOSL) in 2016 in collaboration with the International Agency for the Prevention of Blindness (IAPB) South East Asia revealed the prevalence of blindness in Sri Lanka was 1.6% and 15.4%, respectively, concerning severe visual impairment and blindness. This rate varies for different provinces, where the highest was from Uva and Southern provinces (2.9% and 0.29%) [5]. Additionally, it was found that most of them experience higher financial pressure and various barriers like lack of access to healthcare, insufficient water, poor sanitation facilities and poor living conditions [6].

Exacerbating the above-mentioned conditions, today's global catastrophe, the 'coronavirus (COVID-19) pandemic' has severely affected these individuals making them more vulnerable, putting them at a high exposure to the risk of increased morbidity and mortality while drastically collapsing the individuals' income level [7, 8]. However, in assessing the extent to which VIB influences income, empirical evidence underlying this association is unknown and an under-researched topic that past researchers have not addressed.

The objective of the present study is to fill this empirical gap. In doing so, the study can be considered extensive, as it investigates the impact of different levels of vision and other socio-economic characteristics on the income of visually impaired people.

This research is significant from the existing studies and contributes to the literature in four ways. Firstly, visual impairment and the income level of these victims have become a major issue in the 21st century, in terms of health and economic aspects. In these circumstances, this issue of visual health requires the attention of regulatory authorities such as the government, health sector, policymakers, other organisations and the general public. Secondly, no previous research study has been conducted concerning the area under consideration, addressing the local scenario. According to available information, this study will be the first endeavour of this kind of econometric research study covering the impact of levels of vision and other socio-economic characteristics towards the income of visually impaired and blind persons based in Sri Lanka. Thirdly, according to the World Health Organization (WHO), over 2.2 billion people suffered from visual disorders worldwide; over 90% of them were from low and middle-income countries, which is the main cause of blindness in Asia and Africa [2]. Therefore, after completing this study, the findings can provide valuable insights to the GOSL for formulating strategies and policies. The findings will also benefit the healthcare sector and policymakers aiming to recover from the setback encountered by the COVID-19 pandemic and revive the economy and QOL of visually impaired people. In this regard, the health sector could utilise these data to spread awareness and expand its healthcare facilities regarding this pandemic, especially among individuals with VIB who are badly affected by the crisis.

The rest of the paper is structured as follows; section 2 describes the method. Section 4 examines empirical findings; finally, section 5 asserts the concluding segment with policy implications that align with this study.

## Materials and methods

### Study population and data collection procedure

To conduct phases one and two of the study, a cross-sectional study was conducted on a segment of the Sri Lankan population mainly consisting of the VIB people living in three different rural areas associated with three provinces. These rural locations included *Polpithigama Divisional Secretariat (DS)* in *Kurunegala*, *Siyanethugama model village for VIB people* in *Hambanthota, and DS* in Jaffna. These three locations are located in the *North Western*, *Southern and Northern* provinces, respectively, as shown in Fig 1, drawn using Geographic Information System (GIS) Software.

The participants for the survey were selected through purposive and convenient sampling techniques, thus including persons with various levels of vision. Thus, the latter was determined based on the criteria of "Vision 2020" research conducted to measure the ophthalmic aspect of vision among people in Sri Lanka [9]. From samples selected from each district, a total of 313 participants were proportionately enrolled in the study resulting in 188 from Polpithigama DS in Kurunegala, 64 from the Hambanthota DS and 61 from the Jaffna DS. The socio-economic and demographic characteristics were identified through a well-designed questionnaire covering questions relating to age, gender, marital status, education status, employability, and income level.

In phase three of the study, explicit interviews were held through focus group discussions (FGDs) to capture the factors affecting the QOL of persons with VIB. Eleven individuals (four females and seven females) were purposively selected to represent the VIB community. All these participants were from various socio-economic backgrounds and distinct cadres in the society, including lawyers, undergraduates, academicians, entrepreneurs, development officers and committee members from diverse VIB voluntary organisations. The selected sample includes VIB individuals from different age categories ranging from 20 to 60 years living in rural and urban areas of the country. Their detailed information is not disclosed due to ethical reasons and privacy. Based on their fluency, the FGDs were held online through the Zoom platform using two languages (Sinhala and English). The interviews lasted, on average 3 hours and 40 minutes. All these discussions were recorded with participants' prior consent and manually transcribed for analytical purposes.

**Statistical analysis.** This research consists of three phases; the first phase details the distribution of socio-economic characteristics along with the income of all participants on people with visual impairment and blindness; the second phase incorporates the mediating effect of socio-economic status on income levels of people with VIB; finally, the third phase comprehends the factors that affect the QOL and impact on income disparities. The data file used for the study is presented in S1 Appendix.

Descriptive statistics were utilised to analyse the identified socio-economic characteristics. Ordered probit regression was employed to determine the mediating effect of socio-economic status on income levels. Word Cloud illustrates the factors affecting the QOL.

Overall, the main objective was to determine the mediating effect of socio-economic status on income levels. Therefore, referring to numerous literature, the current study applied the ordered probit model introduced by Chester Ittner Bliss [10, 11]. According to Aitchison and Silvey [12], the ordered probit model's purpose is to generalise more than two outcomes of an ordinal dependent variable. Therefore, with a reasonable assumption, this ordered probit model would be effective concerning validity and reliability for this study.

Before conducting the Ordered probit regression model, it is vital to contemplate three main considerations as to whether the objective fits with the regression model [10, 13,14,15,16,17]. The three main considerations are model specification, model building, and

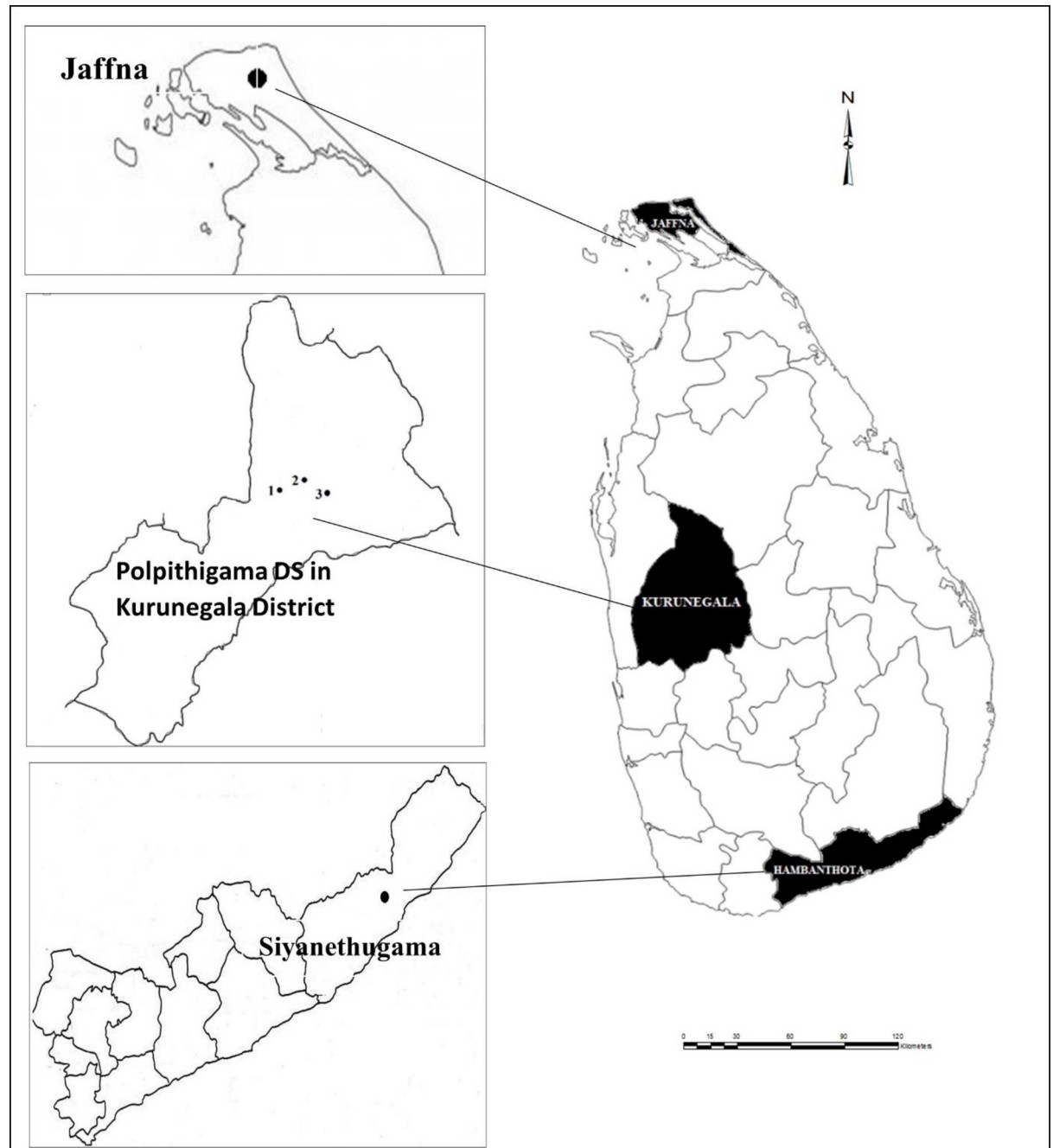

**Fig 1. District allocation of VIB individuals.** Source: Authors' demonstration using GIS.

model diagnosis check. The model specifications allow the researchers to identify the dependent and independent variables identified as income and levels of vision along with other socio-economic factors such as age, gender, marital status, employability, and education. The model building allows the researchers to specify the process to build the equation concerning the variables of the present research study; further, it provides the pathway to explain the variation between income (Dependent variable) and levels of vision along with other socio-economic characteristics (Independent variables). The following equation has been used to

investigate the impact of different levels of vision and other socio-economic characteristics on income.

Based on the study, the authors' general specification of the formula related to the ordered probit model can be expressed as:

$$y_i^* = x_i'\beta + \epsilon_i \tag{1}$$

Here, where $y_i^*$ is a latent variable measure the severity of i[th] individual income. $x_i$ is a (k×1) vector of observed explanatory variables, $\beta$ is a (k×1) vector of the unknown parameters, and $\epsilon_i$ is a random-error term assumed to be normally distributed with a zero mean. The ordered probit model translates the latent variable into observed ordinal-income data of the visually disabled people, y, as follows [18].

$$
\begin{aligned}
&y = 1 \text{ if } y_i^* = \text{no income} \\
&y = 2 \text{ if } y_i^* = \text{income up to SLRs.10,000} \\
&y = 3 \text{ if } y_i^* = \text{income between SLRs.10,000 and SLRs.25,000} \\
&y = 4 \text{ if } y_i^* = \text{income between SLRs.25,000 and SLRs.50,000} \\
&y = 5 \text{ if } y_i^* = \text{income between SLRs.50,000 and SLRs.75,000} \\
&y = 6 \text{ if } y_i^* = \text{income over SLRs.75,000}
\end{aligned}
\tag{2}
$$

The income levels indicated above are derived based on Sri Lanka's Household Income and Expenditure Survey (HIES) 2019. According to the decile groups, the per capita income share is obtained as shown in Table 1.

The marginal effects of the variables obtained from the ordered probit model estimates is used to measure the impact on the probability of income levels due to changes in the various explanatory variables. The marginal effects can be calculated in accordance with [18] as

**Table 1. Definitions of the income levels according to HIES– 2019.**

| Per Capita Income Range (According to HIES 2019) | Share of the household per capita income by socio-economic group | Derivation conducted for this study |
|---|---|---|
| Less than or equal SLRs. 6,222 | Poorest 20% (5.8%) | No income (y = 1) |
| SLRs. 6,223–8,245 | | Income up to SLRs.10,000 (y = 2) |
| SLRs. 8,246–10,114 | Middle 60% (43.6%) | Income between SLRs.10,000 and SLRs.25,000 (y = 3) |
| SLRs. 10,114–11,997 | | |
| SLRs. 11,998–14,095 | | |
| SLRs. 14,096–16,683 | | |
| SLRs. 16,684–20,165 | | |
| SLRs. 20,166–25,502 | | |
| SLRs. 25,503–37,287 | Richest 20% (50.6%) | Income between SLRs.25,000 and SLRs.50,000 (y = 4) |
| | | Income between SLRs.50,000 and SLRs.75,000 (y = 5) |
| More than 37,287 | | Income over SLRs.75,000 (y = 6) |

Source: Authors' compilation based on the Department of Census and Statistics [19].

follows,

$$\partial Prob(Y_i = 1x)/\partial x = -\phi(-x_i\,\beta)\beta$$
$$\partial Prob(Y_i = 2x)/\partial x = \phi(-x_i\beta) - \phi(10,000 - x_i\,\beta)\beta$$
$$\partial Prob(Y_i = 3x)/\partial x = \phi(10,000 - x_i\,\beta) - \phi(25,000 - x_i\,\beta)\beta$$
$$\partial Prob(Y_i = 4x)/\partial x = \phi(25,000 - x_i\,\beta) - \phi(50,000 - x_i\,\beta)\beta$$
$$\partial Prob(Y_i = 5x)/\partial x = \phi(50,000 - x_i\,\beta) - \phi(75,000 - x_i\,\beta)\beta$$
$$\partial Prob(Y_i = 6x)/\partial x = -\phi(75,000 - x_i\,\beta)\beta$$

(3)

where $\phi$ is the standard normal cumulative distribution of $\epsilon_i$. The parameters in the model specified in Eq 3 are estimated using the Maximum Likelihood method. The explanatory variables in Eq 3, such as *age, employability, education, gender, marital status, geographical locations* and *levels of vision*, are summarised in Table 2.

Regarding the independent variables in the current study, the variables include three categories; where the first category encompassed demographics such as *age, gender*, and *marital status*. The second category included socio-economic characteristics such as *education level* and *employment*, further emphasise *education* in line with previous studies as the most commonly used determinants [20,21,22,23]. Lastly, the third category includes the *vision levels* of VIB persons.

In line with mainstream literature, this study considers that the ordered variable is linked to an underlying latent variable in Eq 2, which is divided into six intervals. Thereafter, a forward stepwise regression technique was carried out to select the significant variables. For this purpose, significant variables were selected according to the p value > 0.15; variables more than 0.15 would be removed from the model, whereas those under p value < 0.1 would be retained. The goodness of fit of the model evaluated with overall goodness of fit statistics is computed as follows [24]:

$$\rho^2 = 1 - [L(\beta)/L(c)]$$

(4)

where $L(\boldsymbol{\beta})$ is the maximum log-likelihood value and $L(c)$ is the value of likelihood when it

**Table 2. Definitions of variables with the expected sign.**

| Variable | Description | Expected signs |
|---|---|---|
| **Socio economic characteristics** | | |
| *Age* | Age of the household respondents (in years) | (+) |
| *Employability* | Separate dummy variables for employed, seeking a job, student, household activity, retired, unable to work and others; others are the reference category. | (-/+) |
| *Education* | Separate dummy for no schooling, primary education, secondary education, G. C.E Ordinary (O/L) passed, grade 12–13, Advanced Level (A/L) passed, tertiary and vocational; vocational is the reference category | (+) |
| *Gender* | 1 if male; 0 otherwise | (-/+) |
| *Marital status* | Separate dummy variables for single, married, divorced, widowed; widowed is the reference category. | (-/+) |
| *Geographical locations* | Separate dummy variables for districts of *Hambanthota, Kurunegala* and Jaffna: Jaffna is the reference category. | (-/+) |
| *Levels of vision* | Separate dummy variables for totally blindness, severely impaired, moderately impaired and mild impairment; mild impairment is the reference category. | (+) |

Source: Authors' compilation.

includes only constants. Based on this, the model with the highest goodness of fit value is selected for the analysis. By using ordered probit regression, the study shows significant changes with estimates, robust values, and marginal effects for all *income levels*.

Furthermore, to stimulate and comprehend the factors affecting the QOL influencing income disparities, a WordCloud visualisation tool was employed in this study. Therefore, in accordance with recent empirical studies, past researchers have employed Word Cloud to reveal the qualitative responses in a user-friendly manner. Even with its simple format of illustration to analyse data, many researchers profess it as a righteous qualitative technique [25,26,27,28,29].

**Ethical concerns.** This study was conducted once the Ethical Clearance Board of Sri Lanka Institute of Information Technology (SLIIT) granted ethical clearance approval. Accordingly, the study objectives were conveyed to respondents prior to commencing the survey and their informed consent was obtained. Participants' data are secured with a high level of confidentiality and stored under AHEAD Project (DOR 1 HEMS) for Persons with Visual Impairment and Blindness at the SLIIT Business School, Sri Lanka Institute of Information Technology, Malabe, Sri Lanka.

## Findings and discussion

### Phase 1: The distribution of socio-economic characteristics along with income of all participants on people with visual impairment and blindness

A total of 313 VIB people participated from three purposively selected districts: 64 participants from Hambanthota, 188 from Polpithigama and 61 from Jaffna have been allocated as per this study's responses. The distribution of socio-economic characteristics is displayed in Table 3. The age of participants averaged 44.96 years ranged from >10 to 60+ years and most were married (62.94%). Most participants were males (178–62.30%), out of which 101 were totally blind. Most participants (81) had a remarkably low education level, having passed Ordinary Level (26.47%) and with a monthly income of SLRs. 10,000–25,000 (25.88%). This majority was identified to be employed (42.38%) in diverse working platforms.

### Phase 2: The mediating effect of socio-economic status on income levels of people with vision impairment and blindness

To analyse more specifically on vision levels and socio-economic factors and its impact on income, the study has undergone a (four step) stepwise regression method. As the first step, authors regressed these variables through multiple regression to explore the relevant significance in each income level, as depicted in Table 4. Specifically, according to socio-economic characteristics, in the gender variable, the risk of being a male having visual impairment was not found to be associated with income penetrations. Similarly, regarding a person seeking a job, having passed grades 12 and 13 was also not proved to be associated with income. Furthermore, the marital status of a VIB person being a single, married, or divorced person does not associate with income earnings. As depicted in Table 4, additionally discussing the vision levels, being severely or moderately impaired, neither totally blinded was not associated with income penetrations.

Table 5 demonstrates the third step of the study, which evaluates ordered probit results of the visually incapacitated individual's socio-economic characteristics and the impact towards income. Estimations assert different variations with each income level depicting significance at 1%, 5% and 10% levels, respectively. Furthermore, it reveals the appropriate estimations and how each characteristic significantly describes an impact relevant to the six types of income levels.

**Table 3. Distribution of socio-economic characteristics among all participants with visual impairment and blindness.**

| Variables | Total | Means if(%) | Standard deviation |
|---|---|---|---|
| | N = 313(%) | | |
| **Levels of vision** | | | |
| • Totally blind | 154 (53.10) | | |
| • Severely impaired | 53 (18.28) | | |
| • Moderately impaired | 42 (14.48) | | |
| • Mild impairment | 41 (14.14) | | |
| **Age (years)** | 313 | 44.96 | 19.21 |
| **Gender** | | | |
| • Male | 195 (62.30) | | |
| • Female | 118 (37.70) | | |
| **Marital status** | | | |
| • Single | 98 (31.31) | | |
| • Married | 197 (62.94) | | |
| • Divorced | 8 (2.56) | | |
| • Widowed | 10 (3.19) | | |
| **Employability** | | | |
| • Employed | 128 (42.38) | | |
| • Seeking a job & available to work | 15 (4.97) | | |
| • Student | 71 (23.51) | | |
| • Household activity | 20 (6.62) | | |
| • Retired | 8 (2.65) | | |
| • Unable to work | 30 (9.93) | | |
| • Others | 30 (9.94) | | |
| **Education level** | | | |
| • No schooling | 31 (10.13) | | |
| • Primary education | 80 (26.14) | | |
| • Secondary education | 58 (18.95) | | |
| • G.C.E O/l passed | 81 (26.47) | | |
| • Grade 12–13 | 9 (2.94) | | |
| • Passed A/L | 21 (6.86) | | |
| • Tertiary | 16 (5.23) | | |
| • Vocational | 1 (0.33) | | |
| **Monthly income** | | | |
| • No income | 20 (6.39) | | |
| • Up to 10000 | 75 (23.96) | | |
| • 10000–25000 | 81 (25.88) | | |
| • 25000–50000 | 56 (17.89) | | |
| • 50000–75000 | 23 (7.35) | | |
| • Over 75000 | 4 (1.28) | | |
| **Geographical location** | | | |
| • Hambanthota | 64 (20.45) | | |
| • Kurunegala | 188 (60.06) | | |
| • Jaffna | 61 (19.49) | | |

Source: Author's calculation based on the primary data.

**Table 4. Initial regression results of all variables (Step 1 and 2).**

| Variable | Regression | | Stepwise | |
|---|---|---|---|---|
| | Estimate | Robust SE | Estimate | Robust |
| Constant | -0.1229 | | 0.3129 | |
| **Socio-economic and demographic variables** | | | | |
| Age | -0.0248*** | 0.0050 | -0.0212*** | 0.0038 |
| Male | 0.1697 | 0.1604 | | |
| **Employment status** | | | | |
| Employed | 1.2859*** | 0.2597 | 1.3473*** | 0.2053 |
| Seeking a job | 0.3207 | 0.3588 | | |
| Student | 0.9427*** | 0.2894 | 0.9023*** | 0.2618 |
| Household activity | 1.5667*** | 0.3477 | 1.4960*** | 0.3198 |
| Retired | 3.0186*** | 0.4201 | 2.9957*** | 0.4146 |
| Unable to work | 1.2016*** | 0.3501 | 1.2107*** | 0.3103 |
| **Education** | | | | |
| No schooling | 1.4891*** | 0.4094 | 1.1583*** | 0.3661 |
| Primary education | 1.2838*** | 0.3353 | 1.0490*** | 0.2949 |
| Secondary education | 1.5523*** | 0.3468 | 1.3453*** | 0.3096 |
| G.C.E O/l passed | 1.6798*** | 0.3481 | 1.4684*** | 0.3014 |
| Grade 12–13 | 0.7318 | 0.5809 | | |
| Passed A/L | 2.0249*** | 0.4177 | 1.7253*** | 0.3845 |
| Tertiary | 2.3976*** | 0.3938 | 2.1834*** | 0.3336 |
| **Marital status** | | | | |
| Single | -0.1240 | 0.4467 | | |
| Married | 0.1726 | 0.4049 | | |
| Divorced | 0.0984 | 0.5468 | | |
| **Geographical location** | | | | |
| *Hambanthota* | 1.3457*** | 0.2365 | 1.4346*** | 0.2310 |
| *Kurunegala* | 0.9429*** | 0.2058 | 0.9081*** | 0.1997 |
| **Levels of vision** | | | | |
| Total blindness | 0.2350 | 0.2013 | | |
| Severely impaired | 0.3799 | 0.2427 | | |
| Moderately impaired | -0.0395 | 0.2417 | | |
| F(23, 289) 16.02 | | | 25.53 | |
| Prob>F 0.0000 | | | 0.0000 | |
| R$^2$ 0.4360 | | | 0.4181 | |
| No of observation 313 | | | | |

Note: *** Significant at 1% level

Table 6 represents the final ordered probit model estimations carried forward as the fourth step, where it asserts the estimations using the independent variables. These results are based on the variables selected for the final ordered probit model where the forward stepwise technique was adopted with p-value <0.10 and removal of variables with p-value ≥0.15. As depicted in Table 5, insignificant variables were excluded from the final estimated models. The marginal effects were separately calculated for no income, up to SLRs.10,000, SLRs.10,000–25,000, SLRs.25,000–50,000, SLRs.50,000–75,000 and above SLRs.75,000 where it provides the interpretation of the effects of the independent variables on income. According to estimates of coefficients illustrated in Table 5, these variables have a contrasting (negative and positive)

**Table 5. Initial ordered probit regression results for all variables (Step 3).**

| Variable | Estimate | Robust SE | (y = 1) | (y = 2) | (y = 3) | (y = 4) | (y = 5) | (y = 6) |
|---|---|---|---|---|---|---|---|---|
| **Socio-economic & demographic characteristics** | | | | | | | | |
| Age | -0.0217*** | 0.0042 | 0.0015*** | 0.0034*** | -0.0028*** | -0.0043*** | -0.0014*** | -0.0000 |
| **Marital status** | | | | | | | | |
| Single | 0.0150 | 0.4490 | -0.0010 | -0.0024 | 0.0019 | 0.0030 | 0.0009 | 0.0000 |
| Married | 0.3107 | 0.4154 | -0.0220 | -0.0462 | 0.0430 | 0.0604 | 0.0188 | 0.0009 |
| Divorced | 0.2742 | 0.5673 | -0.0182 | -0.0499 | 0.0250 | 0.0579 | 0.0224 | 0.0014 |
| **Employment levels** | | | | | | | | |
| Employed | 1.2535*** | 0.2600 | -0.0772*** | -0.1935*** | 0.1017*** | 0.2410*** | 0.1096*** | 0.0095 |
| Seeking a job | 0.3294 | 0.3217 | -0.0217 | -0.0607 | 0.0280* | 0.0699 | 0.0278 | 0.0018 |
| Student | 0.9836*** | 0.2810 | -0.0580*** | -0.1808*** | 0.0417* | 0.2016*** | 0.1041** | 0.0097 |
| Household activity | 1.4149*** | 0.3190 | -0.0620*** | -0.2574*** | -0.0788 | 0.2393*** | 0.2263*** | 0.0388 |
| Retired | 3.0333*** | 0.4928 | -0.0665*** | -0.3258*** | -0.3137*** | -0.0199 | 0.3841*** | 0.4498** |
| Unable to work | 1.1231*** | 0.3272 | -0.0574*** | -0.2143*** | -0.0150 | 0.2201*** | 0.1516** | 0.0187 |
| **Education level** | | | | | | | | |
| No schooling | 1.6574*** | 0.4657 | -0.0692*** | -0.2858*** | -0.1033 | 0.2515*** | 0.2736** | 0.0557 |
| Primary education | 1.4267*** | 0.4156 | -0.0769*** | -0.2473*** | 0.0230 | 0.2688*** | 0.1730** | 0.0215 |
| Secondary education | 1.6628*** | 0.4241 | -0.0780*** | -0.2848*** | -0.0472 | 0.2777*** | 0.2437** | 0.0419 |
| G.C.E O/L passed | 1.7841*** | 0.4260 | -0.0881*** | -0.2890*** | -0.0092 | 0.2996*** | 0.2400*** | 0.0392 |
| Grade 12–13 | 0.9960* | 0.5713 | -0.0505** | -0.1946* | -0.0184 | 0.1988** | 0.1363 | 0.0163 |
| G.C.E A/L passed | 2.0926*** | 0.4692 | -0.0701*** | -0.3155*** | -0.1993*** | 0.1970*** | 0.3783*** | 0.1290 |
| Tertiary | 2.4606*** | 0.4641 | -0.0698*** | -0.3259*** | -0.2581*** | 0.1177 | 0.4257*** | 0.2274 |
| Male | 0.1428 | 0.1415 | -0.0101 | -0.0221 | 0.0191 | 0.2827 | 0.0089 | 0.0004 |
| **Geographical location** | | | | | | | | |
| *Hambanthota* | 1.2493*** | 0.2183 | -0.0675*** | -0.2275*** | 0.0140 | 0.2436*** | 0.1535*** | 0.0180 |
| *Kurunegala* | 0.9277*** | 0.1906 | -0.0625*** | -0.1163*** | 0.1295*** | 0.1693*** | 0.0550*** | 0.0032 |
| **Levels of vision** | | | | | | | | |
| Total blindness | 0.2237 | 1.1762 | -0.0157 | -0.0355 | 0.0287 | 0.0446 | 0.0146 | 0.0007 |
| Severely impaired | 0.3868* | 0.2134 | -0.0257* | -0.0697* | 0.0352*** | 0.0813* | 0.3166 | 0.0020 |
| Moderately impaired | -0.0330 | 0.2130 | 0.0023 | 0.00510 | -0.0043 | -0.0065 | -0.0020 | -0.0001 |
| **Ancillary Parameters** | | **Marginal effects after ordered probit** | | | | | | |
| $\hat{y}_1$ | 1.6060** | 0.6760 | 0.0610 | 0.3153 | 0.3428 | 0.1564 | 0.0290 | 0.0010 |
| $\hat{y}_2$ | 1.9073*** | 0.6870 | | | | | | |
| $\hat{y}_3$ | 2.8476*** | 0.7007 | | | | | | |
| $\hat{y}_4$ | 3.8120*** | 0.7177 | | | | | | |
| $\hat{y}_5$ | 4.8004*** | 0.7345 | | | | | | |
| $\hat{y}_6$ | 6.0012*** | 0.7693 | | | | | | |
| Pseudo R$^2$ | | 0.1704 | | | | | | |
| Log likelihood | | -448.29 | | | | | | |
| Number of observations 313 | | | | | | | | |

Note: a)***significant at 1% levels

**significant at 5% levels

*significant at 10% levels

b) 'y' means income. These are categorised as (y = 1: no income, y = 2: up to 10,000, y = 3: 10,000–25,000, y = 4: 25,000–50,000, y = 5: 50,000–75,000, y = 6: above 75,000).

**Table 6. Ordered probit regression results.**

| Variable | Estimate | Robust SE | (y = 1) | (y = 2) | (y = 3) | (y = 4) | (y = 5) | (y = 6) |
|---|---|---|---|---|---|---|---|---|
| **Socio-economic & demographic characteristics** | | | | | | | | |
| Age | -0.0214*** | 0.0038 | 0.0014*** | 0.0033*** | -0.0027*** | -0.0043*** | -0.0013*** | -0.0000 |
| Married | 0.2704* | 0.1581 | -0.0190 | -0.4051* | 0.0367 | 0.0531* | 0.0166 | 0.0008 |
| **Employment levels** | | | | | | | | |
| Employed | 1.1844*** | 0.2156 | -0.0732*** | -0.1839*** | 0.0978*** | 0.2305*** | 0.1023*** | 0.0087 |
| Student | 0.8911*** | 0.2407 | -0.0534*** | -0.1632*** | 0.0437** | 0.1850*** | 0.0909* | 0.0080 |
| Household activity | 1.2547*** | 0.2862 | -0.0585*** | -0.2342*** | -0.0512 | 0.2301*** | 0.1884*** | 0.0215 |
| Retired | 2.9705*** | 0.4679 | -0.0661*** | -0.3230*** | -0.3114*** | -0.0115 | 0.3911*** | 0.4294** |
| Unable to work | 1.0403*** | 0.2944 | -0.0545*** | -0.1990*** | -0.0055 | 0.2089*** | 0.1352** | 0.0158 |
| **Education level** | | | | | | | | |
| No schooling | 1.5867*** | 0.4667 | -0.0677*** | -0.2766*** | -0.0928 | 0.2512*** | 0.2574** | 0.0498 |
| Primary education | 1.3779*** | 0.4118 | -0.0746*** | -0.2391*** | 0.0248 | 0.2631*** | 0.1649** | 0.0201 |
| Secondary education | 1.6259*** | 0.4201 | -0.0766*** | -0.2789*** | -0.0444 | 0.2758*** | 0.2364** | 0.0400 |
| G.C.E O/L passed | 1.7447*** | 0.4225 | -0.0864*** | -0.2833*** | -0.0076 | 0.2974*** | 0.2330*** | 0.0375 |
| Grade 12–13 | 0.9375 | 0.5708 | -0.0485** | -0.1831* | -0.0113 | 0.1905** | 0.1245 | 0.0144 |
| G.C.E A/L passed | 1.9957*** | 0.4646 | -0.0690*** | -0.3080*** | -0.1854** | 0.2087*** | 0.3599*** | 0.1127 |
| Tertiary | 2.4418*** | 0.4547 | -0.0693*** | -0.3232*** | -0.2571 | 0.1192 | 0.4233*** | 0.2249 |
| **Geographical location** | | | | | | | | |
| *Hambanthota* | 1.2810*** | 0.2206 | -0.0682*** | -0.2310*** | 0.0087 | 0.2480*** | 0.1602*** | 0.0196 |
| *Kurunegala* | 0.9220*** | 0.1916 | -0.0616*** | -0.1154*** | 0.1272*** | 0.1694*** | 0.0551*** | 0.0033 |
| **Levels of vision** | | | | | | | | |
| Total blindness | 0.2207 | 0.1480 | -0.0154 | -0.3491 | 0.0279 | 0.0443 | 0.0145 | 0.0008 |
| Severely impaired | 0.3765** | 0.1905 | -0.0249* | -0.0674* | 0.0339*** | 0.0794* | 0.0308 | 0.0020 |
| **Ancillary Parameters** | | Marginal effects after ordered probit | | | | | | |
| $\hat{y}_1$ | 1.3758 | 0.5235 | 0.0607 | 0.3130 | 0.3425 | 0.1580 | 0.0293 | 0.0010 |
| $\hat{y}_2$ | 1.6743 | 0.5332 | | | | | | |
| $\hat{y}_3$ | 2.6079 | 0.5453 | | | | | | |
| $\hat{y}_4$ | 3.5693 | 0.5617 | | | | | | |
| $\hat{y}_5$ | 4.5601 | 0.5847 | | | | | | |
| $\hat{y}_6$ | 5.7562 | 0.6256 | | | | | | |
| Pseudo R$^2$ | 0.1680 | | | | | | | |
| Log likelihood | -449.62803 | | | | | | | |
| Number of observations 313 | | | | | | | | |

effect on the income of visually disabled people. Each income level shows significant variations. However, according to Table 3, mainly the income level of SLRs.10,000–25,000 (25.88%) was the highest earned by visually impaired individuals.

An individual's age is one crucial component contemplated mainly in the visual disorder category. This is evidenced by literature that most people at different stages of their age face visual impairment issues. Lusardi and Mitchell [30] emphasised that young and old people have greater financial literacy and the middle-aged have less financial literacy. This can be further demonstrated with results that emphasise significant income level estimations. Regarding the 'no income', it can be construed that increasing age will increase the probability of ending up with no income and income 'up to SLRs.10,000' under a significant portion of 0.0014 and 0.003 percentage points, respectively. Mohammed M. Abdull, Selvaraj Sivasubramaniam [31] also proved this situation that increasing age was mainly associated with the increasing

prevalence of all conditions that cause blindness. Furthermore, Table 6 depicts the income levels of SLRs.10,000–25,000, SLRs.25,000–50,000 and SLRs.50,000–75,000 lead to a significant estimation where increasing age decreases the probability of earning higher income levels with a proportion of 0.0027, 0.0043, and 0.0013 percentage points, respectively.

Being compatible with other marital status, VIB people in the married category are restricted to earning an income of up to SLRs.10,000 and SLRs.25,000–50,000 with a significant portion of 0.4051 and 0.0531 percentage points, respectively. Enhancing the analysis into employment levels of the visually disabled individuals, namely employed, student, household activity, retired and people unable to work, have various significant levels with each income type. Besides, these individuals are not restricted to earning only up to an income level of 'no income' and income 'up to SLRs.10,000' with a significant portion, as depicted in Table 6. Further, employed persons and student category are likely to earn an income between SLRs.10,000–25,000, SLRs.25,000–50,000 and SLRs.50,000–75,000, with relevant percentage points depicted in Table 6. Moreover, visually incapacitated individuals in the category of household workers are estimated to have a probability of earning an income between SLRs.25,000–50,000 and SLRs.50,000–75,000 comparable with the income of others, with a significant portion of 0.2301 and 0.1884 percentage points, respectively.

Additionally, the income generating ability of retired persons is restricted to earn an income between SLRs.10,000–25,000 but has the possibility to earn an income between SLRs.50,000–75,000 and above SLRs.75,000 with 0.3911 and 0.4294 percentage points, respectively. People with VIB who are unable to work have the possibility to earn an income between SLRs 25,000–50,000 and 50,000–75,000 with percentage points of 0.2089 and 0.1352, respectively. Concluding based on all employment categories considered in the present study, it is evident that each employment level has a significant impact with no income and income up to SLRs.10,000 but not significant with the income level of above SLRs.75,000.

The education level of visually disabled people can be determined as a significant variable which has a far greater impact on income. Variables as no schooling, primary education, secondary education, O/L passed, grade 12–13, A/L passed, and tertiary education levels have a significant variance with income levels. Findings further denote a significant probability among all these education segments to fall on to no income and income up to SLRs.10,000 with percentage points (Table 6). Moreover, for these individuals qualified with G.C.E.O/L, their possibility is tightened, thereby restricting them to earn an income of SLRs.10,000–25,000 with 0.1854 percentage points, respectively, compared to other segments. Except for the tertiary qualified group of visually disabled individuals, for other education groups, a significant probability exists to earn an income of SLRs.25,000–50,000. Except for individuals who have passed A/L, persons with other education qualifications have a possibility to earn an income of SLRs.50,000–75,000 and percentage points, as illustrated in Table 6.

When ascertaining the marginal effect of the geographical location of household units, if the individual is situated in *Hambanthota*, the probability of heading to a no income level is likely to be restricted with a significant percentage point of 0.0682. Moreover, the probability of earning an income up to SLRs.10,000 will be restricted by 0.2310 percentage points. Nevertheless, a probability exists for visually disabled people located in *Hambanthota* to earn income levels of SLRs.25,000–50,000 and SLRs.50,000–75,000 with 0.2480 and 0.1602 percentage points, respectively. If situated in *Polpithigama*, the probability of leading to a no income level and income up to SLRs.10,000 is likely to be restricted with a significant percentage point of 0.0616 and 0.1154, respectively. A probability exists among the *Polpithigama* visually incapacitated people to earn an income of SLRs.10,000–25,000, SLRs.25,000–50,000 and SLRs.50,000–75,000 with 0.1272, 0.1694 and 0.0551 percentage points, correspondingly.

**The mediating effect of different levels of vision towards income concentrations.** Individuals who suffer from total blindness do not indicate significance with any income. This situation signals the society about the income earning vulnerability of the VIB and hence, the importance of earning at least an income of SLRs 10,000 due to their disorder. However, severely impaired persons show a significant variation in levels of income. The results denote that even after being severely impaired, they have the possibility to earn an income without being no income earners. As per the analysis, the majority of the severely impaired community can earn an income of at least up to SLRs.10,000 by 0.0249 and 0.0674 percentage points. A few of them have the probability to earn an income between SLRs.10,000–25,000 and SLRs.25,000–50,000 with 0.0339 and 0.0794 percentage points. As per the authors' observations, the majority who suffer from severe visual impairment disorders earn an income in the range of SLRs.10,000–25,000. It can be concluded that even with such disorders, this community has the probability of earning an income to satisfy their daily needs.

This situation aligns with the existing literature where certain empirical studies explored people's income generating ability shapes their economic position. This is proved by the researcher Lynch, Harper [32] who opined that a positive relationship exists between income and health; it can be explained that even though being an impaired person limits a person from earning a higher income, such people can at least earn a limited income to fulfil their day-to-day requirements.

## Phase 3: The qualitative analysis on the exploration of factors affecting the QOL

In accordance with the above conclusion, the study further comprehended what factors influence the QOL, leading to understanding the major reasons for these income disparities among people with severe vision imparities. The World Health Organization (WHO) defines QOL "as an individual's perception of their position in life in the context of the culture and value systems in which they live and concerning their goals, expectations, standards and concerns" [33, p.3]. It relates to life satisfaction, including a broad range from physical health, family, education, employment, wealth, safety and security to freedom, religious beliefs, and the environment [34]. The qualitative analysis was conducted by visualising keywords with Word Cloud illustration, which applied responses in detail, as shown in Fig 2.

Accordingly, Fig 2 depicts that the factors such as resources, facilities, education, opportunities, income, employment, and government explicate the common belief–i.e. persons with VIB, mainly those severely impaired, face issues associated with their QOL and also highly impacts on their income levels.

For example, these visually incapacitated individuals' dependency on government support is highlighted as a major keyword. This is because they believed that the government's proactive intervention could provide them with a feasible solution to improve their income levels alongside increasing their QOL. Conclusively based on all these keywords, it can be inferred that concepts such as resources, facilities, education, opportunities, income, employment, and government are related to the driving forces of the QOL among the participants.

## Conclusion

Sri Lanka being a much untapped indicator in terms of visually impaired people, has become a significant and growing issue. Within such a conundrum, this unseen territory of the impaired community faces the burden of earning a worthwhile income for their living. This situation is evidently proven through the results of this study. It was discovered that only one-fourth (25%) of VIB people achieve an income level of SLRs 10,000–25,000/- as the highest earned,

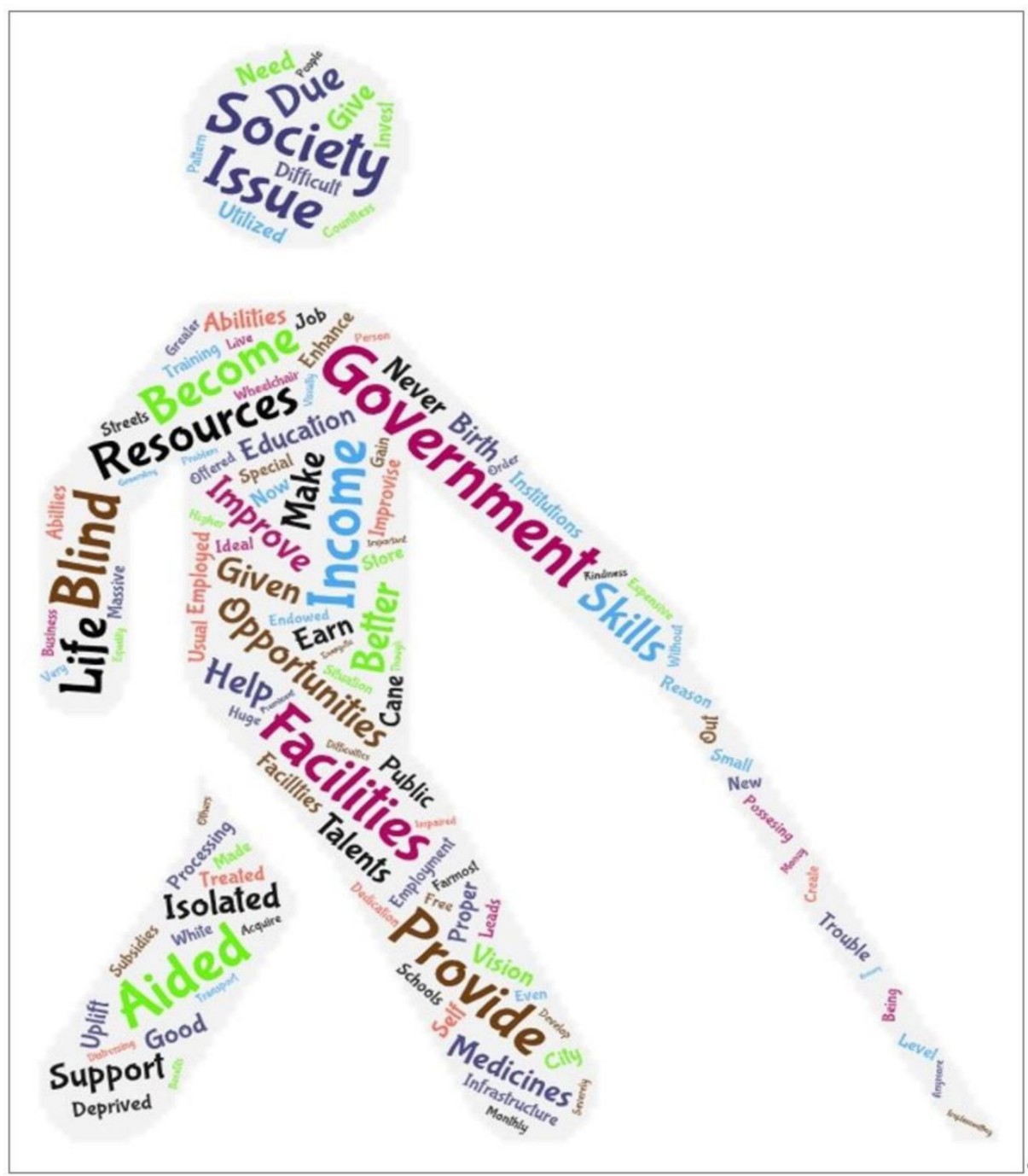

**Fig 2. WordCloud illustration.** Source: Authors' illustration based on qualitative responses.

flagging a serious concerning point. This means the majority (75%) of visually impaired people are vulnerable to lower QOL, poor living conditions, compromised well-being etc. Much complex future issues can stem from this adverse scenario. In this setting, it is recommended to effectively handle the issue to 'nip it from the bud'. Or else, the problem may not be manageable for governments, authorities and policymakers, community and stakeholders (such as sponsors, development organisations).

Allowing VIB people to not only gain financial independence and see the brighter side of their dark world but to strengthen them emotionally and mentally, is the call of the hour. In doing so, it makes them aware that they are not being marginalised, underprivileged or forgotten. Regardless of gender, empowerment of VIB people is recommended for both males and females. E-commerce can be a strong platform for VIB people to market their products, where the physical presence of shops is not required, thus saving cost on rent/cash unnecessarily tied up in large stocks of raw materials of finished goods. This effort needs to be supported through regional coordination; this is because, due to visionary conditions, the VIB community needs support in terms of building their business presence and product promotions/customer relationship management. These include tasks related to computers, smartphones/related devices for frequent product updates that align with approaches of social media/online and e-commerce platforms. Concepts like good market, CAN market etc., in Colombo city, Sri Lanka, can be extended for VIB entrepreneurs/self-employed VIB people to secure markets and help them fetch good prices for their products.

The married category achieves an income level of SLRS 25,0000–50,000. However, as mentioned previously, this portion does not represent the bulk of VIB. Hence, it is vital to enhance the earning potential of VIB, considering their household expenses are relatively high compared to single, divorced or widowed VIB people.

Priority should also be given to VIB people with increasing age. Further, the VIB population ageing, and the rising trend of the ageing population can aggravate several social issues. Income security becomes a burning issue amidst intensifying socio-economic variables. Therefore, Sri Lanka should formulate policies to minimise such risks and social costs.

The present study has some limitations which can be addressed in future studies. The survey was restricted to three highly ranked districts with visually impaired people. Another drawback is that the feedback of some respondents has not been incorporated well enough to provide the most accurate information. Even with these limitations, the study reveals the rising social issue of the inability of this community to generate a worthwhile/reasonable income due to their visual incapacities. Therefore, the proposed policies would be worth catering to this social group to address the barriers/root causes of wealth prosperity.

## Supporting information

**S1 Appendix. Data file.**
(XLSX)

## Acknowledgments

Authors acknowledge the contributions of following institutions and people involved. This is an outcome of research project on 'Quality of Life and Employability potential of Persons with Visual Impairment and Blindness in Sri Lanka' of Business School of the Sri Lanka Institute of Information Technology, Colombo Sri Lanka under the Accelerating Higher Education Expansion and Development (AHEAD) project in Sri Lanka, which is coordinated by the Operations Monitoring and Support Team (OMST) of the Ministry. The authors also would like to thank Ms. Gayendri Karunarathne for proof-reading and editing this manuscript.

## Author Contributions

**Conceptualization:** Colinie Wickramaarachchi, Ruwan Jayathilaka, Theekshana Suraweera.

**Data curation:** Colinie Wickramaarachchi, Lakshika Kollure, Thushya Liyanage, Wageesha Serasinghe, Samanthi Bandara.

**Formal analysis:** Colinie Wickramaarachchi, Ruwan Jayathilaka.

**Funding acquisition:** Theekshana Suraweera, Samantha Thelijjagoda.

**Investigation:** Ruwan Jayathilaka.

**Methodology:** Ruwan Jayathilaka.

**Project administration:** Theekshana Suraweera, Samantha Thelijjagoda.

**Resources:** Samanthi Bandara.

**Software:** Colinie Wickramaarachchi.

**Supervision:** Ruwan Jayathilaka, Theekshana Suraweera, Samantha Thelijjagoda.

**Validation:** Colinie Wickramaarachchi, Ruwan Jayathilaka, Lakshika Kollure, Wageesha Serasinghe.

**Visualization:** Colinie Wickramaarachchi, Thushya Liyanage.

**Writing – original draft:** Colinie Wickramaarachchi, Ruwan Jayathilaka, Theekshana Suraweera.

**Writing – review & editing:** Ruwan Jayathilaka.

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
