## [Decision Letter · Decision Letter 0]

28 Dec 2022

PONE-D-22-15053Can Visual Impairment be a Drain on Your Income Potential?

PLOS ONE

Dear Dr. Ruwan Jayathilaka,

Thank you for submitting your manuscript to PLOS ONE. After careful consideration, we feel that it has merit but does not fully meet PLOS ONE’s publication criteria as it currently stands. Therefore, we invite you to submit a revised version of the manuscript that addresses the points raised during the review process.

We look forward to receiving your revised manuscript.

Kind regards,

Pratap Chandra Mohanty, Ph.D.

Academic Editor

PLOS ONE

https://journals.plos.org/plosone/s/fileid=ba62/PLOSOne_formatting_sample_title_authors_affiliations.pdf.2.

2. During your revisions, please note that a simple title correction is required. Please change the words ' be a drain on' to 'impact': "Can Visual Impairment Impact Your Income Potential?" Please ensure this is updated in the manuscript file and the online submission information.

“Authors acknowledge the contributions of following institutions and people involved. This is an outcome of research project on ‘Quality of Life and Employability potential of Persons with Visual Impairment and Blindness in Sri Lanka’ of Business School of the Sri Lanka Institute of Information Technology, Colombo Sri Lanka.  This project is supported by the World Bank funded Accelerating Higher Education Expansion and Development (AHEAD) programme implemented through the Ministry of Higher Education and the University Grants Commission (UGC) of Sri Lanka, which is coordinated by the Operations Monitoring and Support Team (OMST) of the Ministry. The authors also would like to thank Ms. Gayendri Karunarathne for proof-reading and editing this manuscript.”

“This work was supported by the World Bank, Sri Lanka (AHEAD DOR Grant number 09). The funders had no role in study design, data collection and analysis, decision to publish, or preparation of the manuscript.”

6. We note that you have indicated that data from this study are available upon request. PLOS only allows data to be available upon request if there are legal or ethical restrictions on sharing data publicly. For more information on unacceptable data access restrictions, please see http://journals.plos.org/plosone/s/data-availability#loc-unacceptable-data-access-restrictions.

7. We note that [Figure 1] in your submission contain [map/satellite] images which may be copyrighted. All PLOS content is published under the Creative Commons Attribution License (CC BY 4.0), which means that the manuscript, images, and Supporting Information files will be freely available online, and any third party is permitted to access, download, copy, distribute, and use these materials in any way, even commercially, with proper attribution. For these reasons, we cannot publish previously copyrighted maps or satellite images created using proprietary data, such as Google software (Google Maps, Street View, and Earth). For more information, see our copyright guidelines: http://journals.plos.org/plosone/s/licenses-and-copyright.

Natural Earth (public domain): http://www.naturalearthdata.com/.

Reviewers' comments:

Reviewer's Responses to Questions

**Comments to the Author**

1. Is the manuscript technically sound, and do the data support the conclusions?

Reviewer #1: Partly

Reviewer #2: Partly

2. Has the statistical analysis been performed appropriately and rigorously? 

Reviewer #1: No

Reviewer #2: No

3. Have the authors made all data underlying the findings in their manuscript fully available?

Reviewer #1: Yes

Reviewer #2: Yes

4. Is the manuscript presented in an intelligible fashion and written in standard English?

Reviewer #1: No

Reviewer #2: Yes

5. Review Comments to the Author

Reviewer #1: The authorps present a paper on income and visual impairment/blindness in a sample of adults from Sri Lanka. The following are some comments for consideration and potential improvement of the manuscript:

-There were many points in the manuscript where the language used felt awkward or interferred with my ability to understand the points that the authors were trying to convey. Assistance from an outside editior may be helpful in improving the readability and clarity of the manuscript. For example, I am not sure what the follwing paragraph means:

"From the standpoint of total individuals, it was discovered that persons who suffer

from some sort of visual impairment are more vulnerable to be tagged mainly with earning

income levels of up to SLRs.10,000 and SLRs.10,000-25,000. Remarkably, it was further

identified that most severely impaired persons earn a significant income comparable to the

above-mentioned income levels. It can be concluded that persons with severe vision disorders

have the probability of earning at least up to an income level of up to SLRs.10,000 and

SLRs.10,000 to 25,000 correspondingly."

-In the introduction, more information on prevalence of blindness and visual impairment in Sri Lanka specifically would be appreciated.

-More context for what these income levels mean in the Sri Lankan context (e.g., poverty, lower class, middle class, upper class would be helpful) in contextualizing the results.

-The qualitative portion of the study(Phase 3) was extremely under-developed in its presentation. Qualitative methodology is rich and complex and yet no information was given on the interview content and methods, the coding methods and approaches for the qualitative data, how inter-coder agreement was assessed, how translation occured if needed, etc. This makes it more or less impossible to truly understand the rigour or meaningfulness of the Phase 3 results.

Reviewer #2: I would like to thank all authors for their effort on producing this manuscript. However I have some major concerns.

1.Authors have been mention in the abstract "Due to this excruciating elevated rate on medication and treatments to cure the optical concerns, the visual disorders of people keep rising day by day." I have my doubts on this point. Many different eye conditions can cause low vision, but the most common causes are: Age-related macular degeneration (AMD) , Cataracts.

Diabetic retinopathy (a condition that can cause vision loss in people with diabetes). Its mainly modifiable and non modifiable risk factors associated with NCDs.

2. The objective of this study is to "identify the impact of different levels of vision and socio-economic characteristics towards income of people". I am not very clear about this point weather authors have demonstrated the relevant results. Regression analysis was mainly conducted different Income levels as dependent variable. I am unable to find out confounding effect on socioeconomic characteristics towards impaired vision and income.

3. Authors have mentioned "Majority of severely impaired individuals are more likely to earn a lower rate of income leading them to have a degraded quality of life." It is not clear how quality of life has been measured.

4. To my understanding visual impairment has always direct effect/impact on income and quality of life. The different visual impalement levels could have different levels of quality of life values/levels.Are there any socio-economic characteristics which significantly associated with theses different stages. This can be very important for policy decisions.

5. If possible performing a diamention reduction methods and path analysis will help to assess the hypothesis on above mention points. I believe identifying few unique risk factors which can improved in a developing country with minimal financial input will be more benefited to the community and practically applicable.

6. PLOS authors have the option to publish the peer review history of their article (what does this mean?). If published, this will include your full peer review and any attached files.

Reviewer #1: No

Reviewer #2: No

---

## [Author Response · Author response to Decision Letter 0]

11 Jan 2023

Point by point response to editor and reviewers

Dear editor and the reviewers,

We would like to express our profound appreciation to the editor and the reviewers for the valuable comments and suggestions made on our manuscript which were very helpful in revising and improving it.

Please note that the line numbers referred in this document is aligned with the revised manuscript which has track changes.

Academic Editor Comment 1: Please ensure that your manuscript meets PLOS ONE's style requirements, including those for file naming.

Authors’ Response: Thank you for your comment. The point has been taken into consideration and kindly note that we have followed the guidelines given and adjusted our manuscript accordingly.

Academic Editor Comment 2: During your revisions, please note that a simple title correction is required. Please change the words ' be a drain on' to 'impact': "Can Visual Impairment Impact Your Income Potential?" Please ensure this is updated in the manuscript file and the online submission information.

Authors’ Response: Noted on this comment made, with many thanks! The title is corrected as suggested by the editor in the revised manuscript

Academic Editor Comment 3: We note that the grant information you provided in the ‘Funding Information’ and ‘Financial Disclosure’ sections do not match.

Authors’ Response: Duly noted on the comment with thanks. The funding information/disclosure is now in the same format as follows, 

 “This work was supported by the World Bank, Sri Lanka (AHEAD DOR Grant number 09). The funders had no role in study design, data collection and analysis, decision to publish, or preparation of the manuscript”

Academic Editor Comment 4: Thank you for stating the following in the Acknowledgments Section of your manuscript:

“Authors acknowledge the contributions of following institutions and people involved. This is an outcome of research project on ‘Quality of Life and Employability potential of Persons with Visual Impairment and Blindness in Sri Lanka’ of Business School of the Sri Lanka Institute of Information Technology, Colombo Sri Lanka. This project is supported by the World Bank funded Accelerating Higher Education Expansion and Development (AHEAD) programme implemented through the Ministry of Higher Education and the University Grants Commission (UGC) of Sri Lanka, which is coordinated by the Operations Monitoring and Support Team (OMST) of the Ministry. The authors also would like to thank Ms. Gayendri Karunarathne for proof-reading and editing this manuscript.”

“This work was supported by the World Bank, Sri Lanka (AHEAD DOR Grant number 09). The funders had no role in study design, data collection and analysis, decision to publish, or preparation of the manuscript.”

Authors’ Response: Well noted and many thanks for notifying about this information. The acknowledgement section mentioned in the revised manuscript is updated as follows, 

“Authors acknowledge the contributions of following institutions and people involved. This is an outcome of research project on ‘Quality of Life and Employability potential of Persons with Visual Impairment and Blindness in Sri Lanka’ of Business School of the Sri Lanka Institute of Information Technology, Colombo Sri Lanka under the Accelerating Higher Education Expansion and Development (AHEAD) project in Sri Lanka, which is coordinated by the Operations Monitoring and Support Team (OMST) of the Ministry. The authors also would like to thank Ms. Gayendri Karunarathne for proof-reading and editing this manuscript”.

We have removed all the funding information from the acknowledgement section and other parts of the manuscript as suggested and advised by the editor.

The funding statement will remain as it is. 

Academic Editor Comment 5: In your Data Availability statement, you have not specified where the minimal data set underlying the results described in your manuscript can be found. PLOS defines a study's minimal data set as the underlying data used to reach the conclusions drawn in the manuscript and any additional data required to replicate the reported study findings in their entirety. All PLOS journals require that the minimal data set be made fully available. For more information about our data policy, please see http://journals.plos.org/plosone/s/data-availability.

Authors’ Response: Well noted and many thanks for notifying about this information. The minimal data set of the study will be submitted along with the revised manuscript under S1 Appendix.

Academic Editor Comment 6: We note that you have indicated that data from this study are available upon request. PLOS only allows data to be available upon request if there are legal or ethical restrictions on sharing data publicly. For more information on unacceptable data access restrictions, please see http://journals.plos.org/plosone/s/data-availability#loc-unacceptable-data-access-restrictions.

Authors’ Response: Well noted and many thanks for notifying about this information. The authors have decided to publicly share the data without legal or ethical restrictions. 

Academic Editor Comment 6.1: If there are ethical or legal restrictions on sharing a de-identified data set, please explain them in detail (e.g., data contain potentially sensitive information, data are owned by a third-party organization, etc.) and who has imposed them (e.g., an ethics committee). Please also provide contact information for a data access committee, ethics committee, or other institutional body to which data requests may be sent.

Authors’ Response: Thank you for the information. The authors have decided to share the data set without any hesitation with the journal.

Academic Editor Comment 6.2: If there are no restrictions, please upload the minimal anonymized data set necessary to replicate your study findings as either Supporting Information files or to a stable, public repository and provide us with the relevant URLs, DOIs, or accession numbers. For a list of acceptable repositories, please see http://journals.plos.org/plosone/s/data-availability#loc-recommended-repositories.

Authors’ Response: Well noted. The minimal anonymized data set of the study will be uploaded.

Academic Editor Comment 7: We note that [Figure 1] in your submission contain [map/satellite] images which may be copyrighted. All PLOS content is published under the Creative Commons Attribution License (CC BY 4.0), which means that the manuscript, images, and Supporting Information files will be freely available online, and any third party is permitted to access, download, copy, distribute, and use these materials in any way, even commercially, with proper attribution. For these reasons, we cannot publish previously copyrighted maps or satellite images created using proprietary data, such as Google software (Google Maps, Street View, and Earth). For more information, see our copyright guidelines: http://journals.plos.org/plosone/s/licenses-and-copyright.

Authors’ Response: Thank you very much for indicating this fact. Kindly note that Figure 01 is an author demonstrated diagram through the Geographic Information System Mapping. Therefore, this submission does not contain any copyrighted images. The following statement is added to the manuscript from line number 127 to 129 in page number 7 under the materials and methods section.

…. “These three locations are located in North-western, Southern and Northern provinces, respectively, as shown in Figure 1, drawn using Geographic Information System (GIS) Software”.

Academic Editor Comment 7.1: You may seek permission from the original copyright holder of Figure 1 to publish the content specifically under the CC BY 4.0 license. 

Authors’ Response: Thank you very much and well noted. The response to this comment is given in comment 7.

Academic Editor Comment 7.1: Please upload the completed Content Permission Form or other proof of granted permissions as an ""Other"" file with your submission.

Authors’ Response: Thank you very much and well noted. The response to this comment is given in comment 7.

Reviewer 1 General Comment : The authors present a paper on income and visual impairment/blindness in a sample of adults from Sri Lanka. The following are some comments for consideration and potential improvement of the manuscript:

Authors’ Response: Thank you very much for sparing your valuable time on reviewing this manuscript. The comments given are very well appreciated and they have been incorporated accordingly. 

Reviewer 1 Comment 1: There were many points in the manuscript where the language used felt awkward or interfered with my ability to understand the points that the authors were trying to convey. Assistance from an outside editor may be helpful in improving the readability and clarity of the manuscript. For example, I am not sure what the following paragraph means:

"From the standpoint of total individuals, it was discovered that persons who suffer from some sort of visual impairment are more vulnerable to be tagged mainly with earning income levels of up to SLRs.10,000 and SLRs.10,000-25,000. Remarkably, it was further identified that most severely impaired persons earn a significant income comparable to the above-mentioned income levels. It can be concluded that persons with severe vision disorders have the probability of earning at least up to an income level of up to SLRs.10,000 and SLRs.10,000 to 25,000 correspondingly."

Authors’ Response: Thank you for indicating this point for improvement. The paper has been revised thoroughly and in-depth proofreading check has been performed by a linguistic professional. 

The following paragraph was strengthened with further elaboration in the revised manuscript in page number 25 from line 50 to 54 as follows, 

…….“As per the authors' observations, the majority who suffer from severe visual impairment disorders earn an income in the range of SLRs.10,000-25,000. It can be concluded that even with such disorders, this community has the probability of earning an income to satisfy their daily needs”.

Reviewer 1 Comment 2: In the introduction, more information on prevalence of blindness and visual impairment in Sri Lanka specifically would be appreciated.

Authors’ Response: Thank you very much and the revised version has intensified the introduction. The information on the “prevalence of blindness and visual impairment in Sri Lanka” are stated in the introduction section from line 81 to 85 in page 5

…..“ The survey conducted by the Government of Sri Lanka (GOSL) in 2016 revealed the prevalence of blindness in Sri Lanka was 1.6% and 15.4%, respectively, concerning severe visual impairment and blind conditions. This rate varies for different provinces, where the highest was from Uva and Southern provinces (2.9% and 0.29%)”. 

Reviewer 1 Comment 3: More context for what these income levels mean in the Sri Lankan context (e.g., poverty, lower class, middle class, upper class would be helpful) in contextualizing the results.

Authors’ Response: Thank you for the insightful comment. The methodology section of the paper has been improved by adding the meaning of income levels in the Sri Lankan context. The detail information is available in Table 1. The explanation starts from line 208 to 212 in page 11 and 12.

Reviewer 1 Comment 4: The qualitative portion of the study (Phase 3) was extremely under-developed in its presentation. Qualitative methodology is rich and complex and yet no information was given on the interview content and methods, the coding methods and approaches for the qualitative data, how inter-coder agreement was assessed, how translation occurred if needed, etc. This makes it more or less impossible to truly understand the rigour or meaningfulness of the Phase 3 results.

Authors’ Response: Well noted with many thanks for the comment. The authors agree with the reviewer’s comment and the qualitative portion of the study is now developed with the necessary information as suggested by this reviewer. The detailed explanation is available in the revised manuscript from line 157 to 174 in page 9 as follows, 

“In phase three of the study, explicit interviews were held through focus group discussions (FGDs) to capture the factors affecting the quality of life of persons with VIB. Eleven individuals (four females and seven females) were purposively selected to represent the VIB community. All these participants were from various socio-economic backgrounds and distinct cadres in the society, including lawyers, undergraduates, academicians, entrepreneurs, development officers and committee members from diverse VIB voluntary organisations. The selected sample includes VIB individuals from different age categories ranging from 20 to 60 years living in rural and urban areas of the country. Their detailed information is not disclosed due to ethical reasons and privacy. The FGDs were held online through zoom platform using two languages (Sinhala and English) based on their fluency. The interviews lasted on average 3 hours and 40 minutes. All these discussions were recorded with participants’ prior consent and manually transcribed for analytical purposes”.

Reviewer 2 General Comment : I would like to thank all authors for their effort on producing this manuscript. However, I have some major concerns.

Authors’ Response: Thank you very much for reviewing this paper and we have addressed the valuable comments given in the revised manuscript.

Reviewer 2 Comment 1: Authors have been mentioned in the abstract "Due to this excruciating elevated rate on medication and treatments to cure the optical concerns, the visual disorders of people keep rising day by day." I have my doubts on this point. Many different eye conditions can cause low vision, but the most common causes are: Age-related macular degeneration (AMD), Cataracts.

Diabetic retinopathy (a condition that can cause vision loss in people with diabetes). It’s mainly modifiable and non-modifiable risk factors associated with NCDs.

Authors’ Response: Thank you for the valuable comment! We agree with the point raised by this reviewer. Many eye conditions can cause low vision as you highlighted like AMD, Cataracts and Diabetic retinopathy. 

But even with such reasons one can increase his/her low vision issues, when there is a higher price rate on medication and treatments to cure the optical concerns. 

This was what we wanted to emphasise from that sentence. However, since the reviewer had a doubt, and we amended the sentence and it is available in the revised manuscript from line number 36 to 39 in page number 3 as follows,

“Different eye conditions can cause low vision. Apart from all other reasons, due to the unbearable higher price rates on medication and treatments, people's visual disorders keep rising daily keep rising daily.”

Reviewer 2 Comment 2: The objective of this study is to "identify the impact of different levels of vision and socio-economic characteristics towards income of people". I am not very clear about this point whether authors have demonstrated the relevant results. Regression analysis was mainly conducted different Income levels as dependent variable. I am unable to find out confounding effect on socioeconomic characteristics towards impaired vision and income.

Authors’ Response: Thank you for the comment. As per the objective of the study, we only discovered whether there is an effect from vision impairment and socio-economic characteristics towards income. 

To further clarify, this study does not find the confounding effect on socio-economic characteristics towards impaired vision and income as it goes beyond the objective of this study. 

In the results and discussion section, under phase two section from line 284 to 295 in page number 16 explains the mediating effect of vision levels and socio-economic status on income levels of this community.

Reviewer 2 Comment 3: Authors have mentioned "Majority of severely impaired individuals are more likely to earn a lower rate of income leading them to have a degraded quality of life." It is not clear how quality of life has been measured.

Authors’ Response: Whilst thanking for the comment, we would like to highlight that there is no specific tool to measure the quality-of-life dimension. Since the reviewer has a doubt on the clarity of the quality-of-life dimension, we would like to bring out this valuable comment into consideration. This part is added to the results and discussion section under the phase three in page 26 from line 72 to 77 in the revised manuscript as follows, 

….“The World Health Organization (WHO) defines Quality of Life “as an individual’s perception of their position in life in the context of the culture and value systems in which they live and concerning their goals, expectations, standards and concerns” [28, p.3]. It relates to life satisfaction, including everything from physical health, family, education, employment, wealth, safety, security to freedom, religious beliefs, and the environment…”

Reviewer 2 Comment 4: To my understanding visual impairment has always direct effect/impact on income and quality of life. The different visual impairment levels could have different levels of quality-of-life values/levels. Are there any socio-economic characteristics which significantly associated with these different stages. This can be very important for policy decisions.

Authors’ Response: Thank you for indicating this point for enhancement. As per the objective of the study, we investigated the impact of socio-economic characteristics and vision levels towards income. For this study, we didn’t find the association of how socio-economic characteristics are associated with different impairment stages. 

Therefore, as per the reviewer’s valuable suggestion, to identify how socio-economic characteristics associate with different impairment levels, we will be conducting another study. It is with great thanks; we would like to incorporate this significant comment into consideration for our next study which is currently under the drafting process.

Reviewer 2 Comment 5: If possible, performing a dimension reduction methods and path analysis will help to assess the hypothesis on above mention points. I believe identifying few unique risk factors which can improved in a developing country with minimal financial input will be more benefited to the community and practically applicable.

Authors’ Response: Thank you very much for indicating this point for development. Currently we are in the process of collecting an island wide data collection from the VIB community of Sri Lanka which will be completed by the year 2024. We will be incorporating a larger sample size for the next study and for that we are planning to apply the recommended dimension reduction method and path analysis. 

We highly believe that it can be helpful to identify the unique risk factors for the development of the country.

---

## [Decision Letter · Decision Letter 1]

20 Mar 2023

PONE-D-22-15053R1Can Visual Impairment Impact Your Income Potential?PLOS ONE

Dear Ruwan Jayathilaka,

Thank you for submitting your manuscript to PLOS ONE. After careful consideration, we feel that it has merit but does not fully meet PLOS ONE’s publication criteria as it currently stands. Therefore, we invite you to submit a revised version of the manuscript that addresses the points raised during the review process.

revise and submit the revised version as per the comments of the reviewer 3.

We look forward to receiving your revised manuscript.

Kind regards,

Pratap Chandra Mohanty, Ph.D.

Academic Editor

PLOS ONE

Journal Requirements:

Additional Editor Comments:

revise the paper as per the comments of A.A. Nilanga Nishad (Reviewer 3) and submit the revised version

Reviewers' comments:

Reviewer's Responses to Questions

**Comments to the Author**

1. If the authors have adequately addressed your comments raised in a previous round of review and you feel that this manuscript is now acceptable for publication, you may indicate that here to bypass the “Comments to the Author” section, enter your conflict of interest statement in the “Confidential to Editor” section, and submit your "Accept" recommendation.

Reviewer #2: All comments have been addressed

Reviewer #3: (No Response)

2. Is the manuscript technically sound, and do the data support the conclusions?

Reviewer #2: Yes

Reviewer #3: No

3. Has the statistical analysis been performed appropriately and rigorously? 

Reviewer #2: Yes

Reviewer #3: No

4. Have the authors made all data underlying the findings in their manuscript fully available?

Reviewer #2: Yes

Reviewer #3: No

5. Is the manuscript presented in an intelligible fashion and written in standard English?

Reviewer #2: Yes

Reviewer #3: No

6. Review Comments to the Author

Reviewer #2: I thank all the authors for there responses. Authors have address all the review comments raised by me. There is a significant improvement and and clarity added to the manuscript.

Reviewer #3: This manuscript needs more extensive editing.

Introduction is not clear and gives misleading information, objectives not clearly written, methods not described adequately, trying to highlight the statistics made it less understandable, more importantly no clear statement on Ethical clearance for human subject research...

Do you have evidence to confirm your first sentence in the abstract?

I am not sure about it. On the other hand I am also not sure on which medications you talk about medications which prevent visual impairment and blindness.

Regarding the objective; Everybody know that socio-economic characteristics will adversely affect towards the income of people. So why do you want to study that?

I recommend getting help from a public health specialist and ophthalmologist to write this article and need more concise writing. Please get help from English editing services as well.

This sentence cannot be clearly understood, “The survey conducted by

the Government of Sri Lanka (GOSL) in 2016 revealed the prevalence of blindness in Sri

Lanka was 1.6% and 15.4%, respectively, concerning severe visual impairment and blind

conditions.”

Blind conditions = blindness

What do you want to say here ? “The survey conducted by

the Government of Sri Lanka (GOSL) in 2016 revealed the prevalence of blindness in Sri

Lanka was 1.6% and 15.4%, respectively, concerning severe visual impairment and blind

Conditions”

Have you got the clearance from an ethical review committee as this study involves human subjects? It is not clear in your statements that you have got is. If yes give the reference numbers and details.

The readers are not in a position to understand the statistics. Kindly make it clear and simple.

7. PLOS authors have the option to publish the peer review history of their article (what does this mean?). If published, this will include your full peer review and any attached files.

Reviewer #2: No

Reviewer #3: **Yes: **A.A.N. Nishad

---

## [Author Response · Author response to Decision Letter 1]

2 Apr 2023

Point by point response to reviewers

Dear Reviewers,

We would like to express our profound appreciation to the reviewers for the valuable comments and suggestions made on our manuscript which were very helpful in revising and improving it. Please note that the line numbers referred in this document is aligned with the revised manuscript which has track changes.

Reviewer 1 comment 1: I thank all the authors for there responses. Authors have address all the review comments raised by me. There is a significant improvement and and clarity added to the manuscript.

Authors' Response to Reviewer 1 comment 1: Thank you very much for the comment and we highly appreciate your time and effort contributed to this manuscript to grow into a success.

Reviewer 2 comment 1: This manuscript needs more extensive editing.

Authors' Response to Reviewer 2 comment 1: Thank you very much and well noted for raising this point. The comment is well addressed with the below suggested comments in the revised version.

Reviewer 2 comment 2: Introduction is not clear and gives misleading information, objectives not clearly written, methods not described adequately, trying to highlight the statistics made it less understandable, more importantly no clear statement on Ethical clearance for human subject research...

Authors' Response to Reviewer 2 comment 2: Well noted and thank you for raising these points. 

Since the objectives are not clearly written, we have re-structured and is now stated from line number 95 to 98 as follows, 

“The objective of the present study is to fill this empirical gap. In doing so, the study can be considered extensive, as it investigates the impact of different levels of vision and other socio-economic characteristics on the income of visually impaired people”

We do agree that the methodology is not described adequately in the version one. In the 2nd round of revision, the same comment was raised, and we incorporated, and the methodology is now in a more understandable and clear flow according to the below structure,

1. Study population and data collection process

2. Statistical analysis with in detail instructions of all types of methodologies used in the study

3. Definitions of each variable

4. Ethical statements 

The ethical sentence is further updated (From line number 267 to 276) with the accessed grant numbers to understand the ethical protection of human utilized in the study as follows, 

“This study was conducted once the Ethical Clearance Board of Sri Lanka Institute of Information Technology (SLIIT) granted ethical clearance approval. Accordingly, the study objectives were conveyed to respondents prior to commencing the survey and their informed consent was obtained. Participants’ data are secured with a high level of confidentiality and stored under AHEAD Project (DOR 1 HEMS) for Persons with Visual Impairment and Blindness at the SLIIT Business School, Sri Lanka Institute of Information Technology, Malabe, Sri Lanka”

Reviewer 2 comment 3: Do you have evidence to confirm your first sentence in the abstract?

I am not sure about it. On the other hand I am also not sure on which medications you talk about medications which prevent visual impairment and blindness.

Regarding the objective; Everybody know that socio-economic characteristics will adversely affect towards the income of people. So why do you want to study that?

Authors' Response to Reviewer 2 comment 3: Well noted and many thanks for notifying about this information. Since the first sentence is doubtful, we have changed it to a more globalized setting regarding the persons with visual impairment and blindness from line number 29 to 42 as follows, 

“People’s quality of life (QOL) has been disrupted globally in the wake of the pandemic in recent times. This was mainly due to global economic crises fueled by the coronavirus (COVID – 19) and other related factors. Sri Lanka, too, was facing major social and economic constraints in the period 2021 –2022. Thus, all communities island wide have been economically disturbed. Among others, people with Visual Impairment and Blindness (VIB) have been pushed to severely disadvantageous positions, financially and otherwise.”

However, this study does not communicate about any preventive medications. It previously discussed about the medicine price escalation due to high inflation (Dabare et al., 2014) 

Therefore, in such a condition, if there is any person who have less vision and can be treatable as per the World Health Organization Statistics, will find it difficult to get the treatments and medicines due to the price escalation.

(“Globally, at least 2.2 billion people have a near or distance vision impairment. In at least 1 billion – or almost half – of these cases, vision impairment could have been prevented or has yet to be addressed”)

(https://www.who.int/news-room/fact-sheets/detail/blindness-and-visual-impairment)

(Dabare, P.R.L., Wanigatunge, C.A. & Beneragama, B.H. A national survey on availability, price and affordability of selected essential medicines for non-communicable diseases in Sri Lanka. BMC Public Health 14, 817 (2014). https://doi.org/10.1186/1471-2458-14-817)

As per your second comment, we admit that everybody knows socio-economic characteristics will adversely affect towards the income of people. Yet, the effect of how persons with visual impairment grapple with the above condition is unknown which led us to conduct this study. Furthermore, there are multiple significances for conducting this kind of a study as follows, 

1. It can enlighten the growth of personalized assistance and training programs that deliver the special needs of visually impaired individuals.

2. This study can offer valuable knowledge to employers and policy makers about the economic benefits of hiring and supporting visually impaired workers.

3. It can help to identify and address discrepancies in income and employment prospects between visually impaired individuals and their sighted peers.

4. The study can expand research into the broader societal influences of visual impairment and the significance of addressing this issue from an economic, as well as a social, perspective.

5. This study can further support to challenge stereotypes and misconceptions about the visually impaired individual’s abilities and potentials in the workplace and society more broadly.

As per your valuable comment we have strengthen the significance of the study from line number 99 to 119 follows, 

“This research is significant from the existing studies and contributes to the literature in four ways. Firstly, visual impairment and the income level of these victims have become a major issue in the 21st century, in terms of health and economic aspects. In these circumstances, this issue of visual health requires the attention of regulatory authorities such as the government, health sector, policymakers, other organisations and the general public. Secondly, no previous research study has been conducted concerning the area under consideration, addressing the local scenario. According to available information, this study will be the first endeavour of this kind of econometric research study covering the impact of levels of vision and other socio-economic characteristics towards the income of visually impaired and blind persons based in Sri Lanka. Thirdly, according to the World Health Organization (WHO), over 2.2 billion people suffered from visual disorders worldwide; over 90% of them were from low and middle-income countries, which is the main cause of blindness in Asia and Africa [2]. Therefore, after completing this study, the findings can provide valuable insights to the GOSL for formulating strategies and policies. The findings will also benefit the healthcare sector and policymakers aiming to recover from the setback encountered by the COVID-19 pandemic and revive the economy and QOL of visually impaired people. In this regard, the health sector could utilise these data to spread awareness and expand its healthcare facilities regarding this pandemic, especially among individuals with VIB who are badly affected by the crisis.”

Reviewer 2 comment 4: I recommend getting help from a public health specialist and ophthalmologist to write this article and need more concise writing. Please get help from English editing services as well.

Authors' Response to Reviewer 2 comment 4: Noted with many thanks for these two valuable suggestions! 

This study comes as a sub project of the AHEAD research project for persons with visual impairment and blindness. Therefore, prior to writing this article, the data collection questionnaires for quantitative and qualitative investigations, was reviewed and finalized, with medical consultations and ophthalmologists and also the study data was further validated by an entire team consisting different specialist from different areas.

Furthermore, for the second comment, the authors would like to appreciate your valuable suggestion. The paper has been revised thoroughly and in-depth proofreading check has been performed by a linguistic professional. 

Reviewer 2 comment 5: This sentence cannot be clearly understood, “The survey conducted by

the Government of Sri Lanka (GOSL) in 2016 revealed the prevalence of blindness in Sri

Lanka was 1.6% and 15.4%, respectively, concerning severe visual impairment and blind

conditions.”

Authors' Response to Reviewer 2 comment 5: Thank you very much for indicating this point for improvement. The following sentence was used to strengthened with further elaboration in the revised manuscript in page number 04 from line 77 to 82 as follows,

“The survey conducted by the VISION-2020 national workshop programme by the Government of Sri Lanka (GOSL) in 2016 in collaboration with the International Agency for the Prevention of Blindness (IAPB) South East Asia revealed the prevalence of blindness in Sri Lanka was 1.6% and 15.4%, respectively, concerning severe visual impairment and blindness”.

Reviewer 2 comment 6: Blind conditions = blindness

Authors' Response to Reviewer 2 comment 6: Duly noted and thank you very much for the suggestion. The suggested comment was well addressed in the revised manuscript in page 4 in line number 82 as follows,

“The survey conducted by the VISION-2020 national workshop programme by the Government of Sri Lanka (GOSL) in 2016 in collaboration with the International Agency for the Prevention of Blindness (IAPB) South East Asia revealed the prevalence of blindness in Sri Lanka was 1.6% and 15.4%, respectively, concerning severe visual impairment and blindness”.

Reviewer 2 comment 7: What do you want to say here ? “The survey conducted by

the Government of Sri Lanka (GOSL) in 2016 revealed the prevalence of blindness in Sri

Lanka was 1.6% and 15.4%, respectively, concerning severe visual impairment and blind

Conditions”

Authors' Response to Reviewer 2 comment 7: Noted with thanks! As per the suggestion given in the second revision by reviewer 01, we included the recent survey statistics of persons with visual impairment and blindness in the Sri Lankan perspective conducted by the vision 2020 program to show the prevalence of these individuals. That is the reason behind including this statement. However, since the statement had some confusions in the previous comment, we revised it and is shown in the above comments 05 and 06.

Reviewer 2 comment 8: Have you got the clearance from an ethical review committee as this study involves human subjects? It is not clear in your statements that you have got is. If yes give the reference numbers and details.

Authors' Response to Reviewer 2 comment 8: Well noted and thank you very much for the comment. This research paper is based on a specific study under the World Bank funded comprehensive research project entitled quality of life and employability potential of persons with visual impairment and blindness in Sri Lanka. The ethical clearance application was submitted to the Dean – Postgraduate Studies and Research and Chairman of the Ethics Committee of Sri Lanka Institute of Information Technology on 16th March 2021. The approval of the ethics committee was obtained on December 5th, 2021. Accordingly, all research activities were conducted strictly under ethical guidelines. This ethical clearance refers to quantitative and qualitative data collections from the persons who are visually challenged and blind in Sri Lanka. 

The ethical clearance paragraph is further updated (line number 267 to 276) in page number 14 as follows, 

“This study was conducted once the Ethical Clearance Board of Sri Lanka Institute of Information Technology (SLIIT) granted ethical clearance approval. Accordingly, the study objectives were conveyed to respondents prior to commencing the survey and their informed consent was obtained. Participants’ data are secured with a high level of confidentiality and stored under AHEAD Project (DOR 1 HEMS) for Persons with Visual Impairment and Blindness at the SLIIT Business School, Sri Lanka Institute of Information Technology, Malabe, Sri Lanka”

Reviewer 2 comment 9: The readers are not in a position to understand the statistics. Kindly make it clear and simple.

Authors' Response to Reviewer 2 comment 9: Thank you for raising this information. As per your valuable suggestion, in order to make the statistics more precise and simpler, the authors incorporated detailed information about the methodology used to investigate the impact of the variables in the revised manuscript and we also incorporated textbooks and different articles to further understand the statistics 

in page number 10 from line 193 to 204 as follows,

“Before conducting the Ordered probit regression model, it is vital to contemplate three main considerations as to whether the objective fits with the regression model [10, 13-17]. The three main considerations are model specification, model building, and model diagnosis check. The model specifications allow the researchers to identify the dependent and independent variables identified as income and levels of vision along with other socio-economic factors such as age, gender, marital status, employability, and education. The model building allows the researchers to specify the process to build the equation concerning the variables of the present research study; further, it provides the pathway to explain the variation between income (Dependent variable) and levels of vision along with other socio-economic characteristics (Independent variables)….”

---

## [Editor Report · Decision Letter 2]

4 Apr 2023

Can Visual Impairment Impact Your Income Potential?

PONE-D-22-15053R2

Dear Dr. Ruwan Jayathilaka,

We’re pleased to inform you that your manuscript has been judged scientifically suitable for publication and will be formally accepted for publication once it meets all outstanding technical requirements.

Kind regards,

Pratap Chandra Mohanty, Ph.D.

Academic Editor

PLOS ONE
---

## [Editor Report · Acceptance letter]

11 Apr 2023

PONE-D-22-15053R2 

Can Visual Impairment Impact Your Income Potential? 

Dear Dr. Jayathilaka:

I'm pleased to inform you that your manuscript has been deemed suitable for publication in PLOS ONE. Congratulations! Your manuscript is now with our production department. 

Kind regards, 

on behalf of

Dr. Pratap Chandra Mohanty 

Academic Editor

PLOS ONE